# Recombinant Modified Vaccinia Virus Ankara Expressing a Glycosylation Mutant of Dengue Virus NS1 Induces Specific Antibody and T-Cell Responses in Mice

**DOI:** 10.3390/vaccines11040714

**Published:** 2023-03-23

**Authors:** Lucas Wilken, Sonja Stelz, Ayse Agac, Gerd Sutter, Chittappen Kandiyil Prajeeth, Guus F. Rimmelzwaan

**Affiliations:** 1Research Center for Emerging Infections and Zoonoses (RIZ), University of Veterinary Medicine (TiHo), 30559 Hannover, Germany; 2Division of Virology, Institute for Infectious Diseases and Zoonoses, Department of Veterinary Sciences, Ludwig Maximilian University (LMU), 80539 Munich, Germany

**Keywords:** dengue virus, vaccine, vector, antibody, T cell

## Abstract

The four serotypes of dengue virus (DENV1–4) continue to pose a major public health threat. The first licenced dengue vaccine, which expresses the surface proteins of DENV1–4, has performed poorly in immunologically naïve individuals, sensitising them to antibody-enhanced dengue disease. DENV non-structural protein 1 (NS1) can directly induce vascular leakage, the hallmark of severe dengue disease, which is blocked by NS1-specific antibodies, making it an attractive target for vaccine development. However, the intrinsic ability of NS1 to trigger vascular leakage is a potential drawback of its use as a vaccine antigen. Here, we modified DENV2 NS1 by mutating an N-linked glycosylation site associated with NS1-induced endothelial hyperpermeability and used modified vaccinia virus Ankara (MVA) as a vector for its delivery. The resulting construct, rMVA-D2-NS1-N207Q, displayed high genetic stability and drove efficient secretion of NS1-N207Q from infected cells. Secreted NS1-N207Q was composed of dimers and lacked N-linked glycosylation at position 207. Prime–boost immunisation of C57BL/6J mice induced high levels of NS1-specific antibodies binding various conformations of NS1 and elicited NS1-specific CD4^+^ T-cell responses. Our findings support rMVA-D2-NS1-N207Q as a promising and potentially safer alternative to existing NS1-based vaccine candidates, warranting further pre-clinical testing in a relevant mouse model of DENV infection.

## 1. Introduction

Dengue virus (DENV) is a mosquito-borne flavivirus closely related to Zika virus (ZIKV), West Nile virus (WNV), and yellow fever virus (YFV) that co-circulates as four antigenically related serotypes (i.e., DENV1–4) and causes nearly 400 million infections each year [1]. About a quarter of all infections are clinically apparent, ranging from dengue fever (DF), a mild, flu-like illness, to the severe dengue haemorrhagic fever (DHF), characterised by thrombocytopenia, coagulopathy, and vascular leakage, that further develops into the life-threatening dengue shock syndrome (DSS) if not immediately treated [2]. Severe dengue rarely develops during primary infection with one serotype, but more frequently occurs during secondary infection with a different (heterologous) serotype [3,4]. This has been largely attributed to the phenomenon of antibody-dependent enhancement (ADE), in which sub-neutralising levels of cross-reactive antibodies to the viral surface proteins precursor membrane (prM) and envelope (E) promote infection of Fc receptor–bearing cells, resulting in increased viral replication and elevated production of pro-inflammatory cytokines [5,6,7,8,9]. The first licenced dengue vaccine, CYD-TDV (Dengvaxia), which expresses the prM and E genes of the four DENV serotypes from the YFV-17D genetic backbone, was partially efficacious in individuals who had previously been infected with DENV, but sensitised seronegative vaccine recipients to enhanced dengue disease [10,11], presumably by mimicking primary infection in these people [12,13]. Another dengue vaccine, TAK-003 (Qdenga), which employs the same chimeric strategy as CYD-TDV but uses a DENV2 genetic backbone, has recently obtained approval in Indonesia, the EU, and the UK. In addition, efforts are ongoing to develop next-generation vaccines that are safe and effective in all populations [14].

Recent insights into the role of DENV non-structural protein 1 (NS1) in viral immune evasion and the pathogenesis of severe dengue disease have attracted interest in its use as a subunit vaccine. Following translation of the viral polyprotein at the rough endoplasmic reticulum (ER) membrane, NS1 monomers are translocated into the ER lumen, where addition of high-mannose glycans to two N-linked glycosylation sites (N130 and N207) occurs [15], followed by homodimerisation. Three distinct domains constitute each monomer: the β-roll domain (amino acids 1 to 29); the wing domain (aa 30–180), which is composed of the α/β subdomain (aa 38–151) and connector subdomains (aa 30–37 and 152–180); and the β-ladder domain (aa 181–352) [16]. NS1 dimers are targeted to sites of viral replication in the ER or transit through the *trans*-Golgi network (TGN), where the N-linked glycan at position 130 is converted to a complex form [17], and are displayed on the plasma membrane [18,19]. In addition, NS1 dimers assemble into hexameric lipoprotein particles that are secreted into the extracellular milieu [20,21,22]. High plasma levels of secreted NS1 have been shown to correlate with the development of severe dengue disease [23]. Several recent reports have identified secreted NS1 as a mediator of vascular leakage, demonstrating that the protein induces endothelial hyperpermeability by triggering the release of vasoactive molecules from immune cells [24,25] or by binding to and being internalised by endothelial cells [24,26,27,28]. Both serotype-specific and cross-reactive NS1-specific antibodies—passively transferred or induced by vaccination—prevent NS1-induced endothelial hyperpermeability and protect mice from lethal DENV infection [24,29,30,31,32], though this seems to depend on the virus strain used in these studies [33]. Moreover, the absence of NS1 from viral particles avoids the risk of ADE, making it a promising candidate antigen for dengue vaccine development. However, since administration of NS1 alone has been shown to trigger vascular leakage and cause morbidity in vivo [24,34], high levels of circulating NS1 following vaccination might have similar consequences, thus potentially posing a safety risk.

Modified vaccinia virus Ankara (MVA) is a promising vector for the delivery of viral antigens, owing to its highly attenuated phenotype and intrinsic immunostimulatory properties [35], and has been successfully used by us and others to generate candidate vaccines against multiple viruses, including influenza viruses, Middle East respiratory syndrome coronavirus (MERS-CoV), and severe acute respiratory syndrome coronavirus 2 (SARS-CoV-2) [36,37,38]. Here, we describe the construction, in vitro characterisation, and immunogenicity of a recombinant MVA expressing a DENV2 NS1 mutant that lacks the N-linked glycosylation site at position 207, which is known to be a determinant of NS1-induced endothelial hyperpermeability [28].

## 2. Materials and Methods

### 2.1. Ethics Statement

Animals were housed and experiments were carried out in strict compliance with European guidelines (EU directive on animal testing 2010/63/EU) and the German Animal Welfare Act. The protocol was approved by the Lower Saxony State Office for Consumer Protection and Food Safety (LAVES, Oldenburg, Germany—33.9-42502-04-21/3806).

### 2.2. Mice

Female C57BL/6J mice were acquired from Charles River Laboratories (Göttingen, Germany) and were used at 6 weeks of age. Animals were housed at the Research Center for Emerging Infections and Zoonoses of the University of Veterinary Medicine, Hannover, Germany under specific-pathogen-free (SPF) conditions in individually ventilated cage (IVC) systems and had access to food and water ad libitum.

### 2.3. Cells and Viruses

Primary chicken embryo fibroblasts (CEFs) were prepared from 10-day-old SPF embryonated chicken eggs (VALO BioMedia, Osterholz-Scharmbeck, Germany), passaged once before use, and cultured in Eagle’s minimum essential medium (EMEM; Sigma-Aldrich, St. Louis, MO, USA) supplemented with 10% heat-inactivated foetal bovine serum (FBS; Gibco, Waltham, MA, USA), 1% penicillin-streptomycin (P/S; Sigma-Aldrich, St. Louis, MO, USA), and 1% non-essential amino acids (NEAAs; Sigma-Aldrich, St. Louis, MO, USA) at 37 °C and 5% CO_2_.

A549 cells (ATCC CCL-185) were maintained in Ham’s F-12K Nutrient Mixture (Gibco, Waltham, MA, USA) supplemented with 10% FBS, 1% P/S (Gibco, Waltham, MA, USA), and 1% GlutaMAX (Gibco, Waltham, MA, USA). HeLa cells (ATCC CCL-2) were cultured in Dulbecco’s modified Eagle’s medium (Gibco, Waltham, MA, USA) supplemented with 10% FBS, 1% P/S, 1% NEAAs (Gibco, Waltham, MA, USA), and 1% GlutaMAX. Both cell lines were kept at 37 °C and 5% CO_2_.

Non-recombinant, wild-type MVA (wtMVA; clonal isolate F6) [35,39] and recombinant MVA expressing green fluorescent protein (GFP) under control of the vaccinia virus (VACV) late promoter P11 (rMVA-GFP) [40] (provided by the Institute for Infectious Diseases and Zoonoses, LMU Munich, Munich, Germany) were propagated in CEFs.

### 2.4. Plasmid Construction

Sequences encoding the second transmembrane domain (TM2) of the E protein, which serves as a natural signal sequence for NS1 [41], and full-length NS1 were extracted from the genome of DENV2 strain New Guinea C (NGC; GenBank accession no. KM204118) and modified as follows: (i) codon 207 of the NS1 coding sequence was modified to encode glutamine (Q) instead of asparagine (N) (i.e., N207Q); (ii) synonymous mutations were introduced to interrupt homonucleotide runs (CCCC and GGGG) and poxvirus early transcription termination signals (TTTTTNT) [42]; (iii) a Kozak sequence was added upstream of the sequence; (iv) a stop codon (TAA) was added downstream of the sequence; and (v) BamHI and PmeI restriction sites were placed at the 5′ and 3′ ends of the sequence, respectively. The resulting construct was synthesised (GenScript, Rijswijk, The Netherlands) and subcloned into the vector plasmid pIIIH5redK1L [43] (provided by the Institute for Infectious Diseases and Zoonoses, LMU Munich, Germany), thereby placing it under transcriptional control of the synthetic VACV early/late PmH5 promoter [44].

### 2.5. Generation of Recombinant MVA

CEFs were seeded at a density of 5 × 10^5^ cells per well into a flat-bottom 6-well plate (Corning, Corning, NY, USA). The next day, the cells were inoculated with rMVA-GFP at a multiplicity of infection (MOI) of 0.05 and incubated for 1 h at 37 °C. Meanwhile, 1 µg of pIIIH5redK1L-D2-NS1-N207Q was mixed with 4 µL of X-tremeGENE HP DNA transfection reagent (Roche, Basel, Switzerland) in 100 µL of serum-free EMEM and incubated for 15 min at RT. The virus inoculum was replaced with infection medium (i.e., growth medium with 2% FBS) and the transfection mix was added dropwise. The cells were then further incubated for 2 days at 37 °C. Red fluorescent (mCherry-positive) foci were picked and plaque-purified until the cultures were devoid of green fluorescent (GFP-positive) foci. Subsequently, non-fluorescent (mCherry-negative) foci were picked and plaque-purified until the cultures were devoid of red fluorescent (mCherry-positive) foci. Primary virus stocks were then prepared by amplification in CEFs and stored at −20 °C.

To confirm correct insertion of the transgene into deletion site III (del III) of the MVA genome, viral genomic DNA was extracted from the rMVA-D2-NS1-N207Q stock using the NucleoSpin Blood QuickPure kit (Macherey-Nagel, Düren, Germany) and analysed by PCR using GoTaq Master Mix (Promega, Fitchburg, WI, USA) along with the del III–specific primers MVA-III-5′ (5′-GAATGCACATACATAAGTACCGGCATCTCTAGCAGT-3′) and MVA-III-3′ (5′-CACCAGCGTCTACAT GACGAGCTTCCGAGTTCC-3′) [43]. Viral genomic DNA of wtMVA was used as a reference. To assess the integrity of the transgene, the PCR product was gel-purified and sequenced externally (Microsynth Seqlab, Göttingen, Germany).

Vaccine preparations were generated by amplification of the primary stock viruses in CEFs and concentration by ultracentrifugation (30,000× *g*, 2 h, 4 °C) through 36% (*w*/*v*) sucrose cushions. Virus pellets were resuspended in 120 mM NaCl 10 mM Tris–HCl (pH 7.4), aliquoted, and stored at −80 °C. Viruses were titrated by immuno-plaque assay using rabbit polyclonal anti-VACV (Lister strain) antibody (OriGene, Rockville, MD, USA; 1:2000), as described previously [43].

### 2.6. Immunofluorescence Staining

Cells were fixed with 4% paraformaldehyde (PFA) in phosphate-buffered saline (PBS) for 15 min at RT and then permeabilised with 0.5% Triton X-100 (Sigma-Aldrich, St. Louis, MO, USA) in PBS for 10 min at RT. Cells were blocked with 2.5% normal horse serum (NHS; Cytiva, Marlborough, MA, USA) in PBS for 30 min at RT, followed by staining with mouse monoclonal anti-DENV NS1 antibody (1:1000; clone FE8, The Native Antigen Company, Oxford, UK) and rabbit polyclonal anti-calnexin antibody (1:500; Sigma-Aldrich, St. Louis, MO, USA) for 1 h at RT. Afterwards, Alexa Fluor 488–conjugated donkey anti-mouse IgG antibody (1:500; Invitrogen, Waltham, MA, USA) and Alexa Fluor 594–conjugated donkey anti-rabbit IgG antibody (1:500; Invitrogen, Waltham, MA, USA) along with NucBlue Live ReadyProbes Reagent (Hoechst 33342, Invitrogen, Waltham, MA, USA) were added for 1 h at RT in the dark. All antibodies were diluted in 2.5% NHS/PBS and cells were washed three times with PBS after each antibody incubation. Images were captured using a Leica DMi8 microscope coupled to a Leica DFC3000 G camera and processed using Leica Application Suite X (LAS X; all Leica Microsystems, Wetzlar, Germany).

### 2.7. Western Blot Analysis

Culture supernatants were collected and clarified by centrifugation at 1000× *g* for 5 min at 4 °C and subsequently kept on ice. Cells were rinsed once with cold DPBS and then harvested on ice in lysis buffer (50 mM Tris, 150 mM NaCl, 1% Triton X-100, 0.5% sodium deoxycholate, 0.1% sodium dodecyl sulphate [SDS], pH 8.0) containing 1X Halt protease and phosphatase inhibitor cocktail (Thermo Fisher Scientific, Waltham, MA, USA). Supernatant samples and whole-cell lysates were resolved by electrophoresis in 10% SDS–polyacrylamide gels under reducing or non-reducing conditions (as specified in the figure legends), and subsequently transferred onto polyvinylidene difluoride (PVDF) Hybond-P membranes (Amersham, Little Chalfont, UK). Membranes were blocked with 5% skimmed milk in Tris-buffered saline containing 0.1% Tween-20 (TBST) for 1 h at RT. Blots were probed with mouse monoclonal anti-DENV NS1 antibody (1:1000; clone FE8, The Native Antigen Company, Oxford, UK) and rabbit monoclonal anti-vinculin antibody (1:5000; clone 3M13, Sigma-Aldrich, St. Louis, MO, USA) diluted in 5% bovine serum albumin (BSA)/TBST. Membranes were washed thrice with TBST and then incubated with horseradish peroxidase (HRP)–conjugated goat anti-mouse IgG antibody (1:5000; Invitrogen, Waltham, MA, USA) or HRP-conjugated goat anti-rabbit IgG antibody (1:5000; Abcam, Cambridge, UK) diluted in 5% skimmed milk/TBST for 1 h at RT. After three washes with TBST, blots were developed using SuperSignal West Pico PLUS chemiluminescent substrate (Thermo Fisher Scientific, Waltham, MA, USA) and subsequently imaged using a ChemiDoc MP imaging system (Bio-Rad, Hercules, CA, USA).

### 2.8. Glycosidase Treatment

The glycosylation status of secreted NS1 was assessed by digesting supernatant samples with peptide:N-glycosidase F (PNGase F; New England Biolabs, Ipswich, MA, USA) or endoglycosidase H (Endo H; New England Biolabs, Ipswich, MA, USA) according to the manufacturer’s instructions and subsequent Western blot analysis as described above.

### 2.9. Peptides

A peptide array spanning the NS1 protein of DENV2 strain NGC (NR-508) was obtained from BEI Resources (NIAID, NIH, Bethesda, MD, USA) and consisted of 15- to 19-mers with 10 or 11 amino acid overlaps. Peptides were dissolved in DMSO or water and pooled as indicated in Appendix A Table A1. Peptide pools were further diluted in water to a concentration of 80 µg/mL per peptide, aliquoted, and stored at −80 °C until use.

### 2.10. Mouse Immunisations

Groups of 6-week-old, female C57BL/6J mice (*n* = 4) were immunised twice intramuscularly (days 0 and 28) with 10^7^ PFU of wtMVA or rMVA-D2-NS1-N207Q. Blood was withdrawn on days 0 (pre-immune), 28 (pre-boost), and 56 (post-boost), and allowed to clot. Serum was subsequently separated by centrifugation in MiniCollect Z Serum Sep tubes (Greiner Bio-One, Kremsmünster, Austria) at 3000× *g* for 10 min at RT, aliquoted, and subsequently stored at −20 °C until use. Spleens were harvested on day 56, processed into single-cell suspensions using the gentleMACS Octo Dissociator (Miltenyi Biotec, Bergisch Gladbach, Germany) and successively passed through 100 µm and 70 µm cell strainers (Miltenyi Biotec, Bergisch Gladbach, Germany). Erythrocytes were lysed in ACK lysing buffer (Gibco, Waltham, MA, USA) for 5 min at RT, followed by washing with cold PBS containing 2% FBS. Splenocytes were eventually resuspended in RPMI 1640 (Gibco, Waltham, MA, USA) supplemented with 10% FBS, 10 mM HEPES, 1% P/S, and 5 µM β-mercaptoethanol (Sigma-Aldrich, St. Louis, MO, USA) (R10F) and kept on ice until further use.

### 2.11. ELISA

MaxiSorp flat-bottom 96-well plates (Nunc, Roskilde, Denmark) were coated with recombinant hexameric NS1 of DENV2 strain 16681 (1 µg/mL; Bio-Rad, Hercules, CA, USA) in 0.1 M carbonate/bicarbonate buffer (pH 9.6) overnight at 4 °C. To investigate the effects of NS1 conformation on antibody binding, additional plates were coated in the same manner with NS1 treated with 0.1% SDS (i.e., dimer) [45] or NS1 treated with 0.1% SDS and heated for 5 min at 95 °C (i.e., monomer) [15]. The next day, plates were washed thrice with PBS containing 0.05% Tween-20 (PBST) and blocked with 2% BSA/PBST for 1 h at 37 °C. Sera were heat-inactivated for 30 min at 56 °C, diluted 100-fold in 2% BSA/PBST, then added to the wells and incubated for 1 h at 37 °C. Plates were washed thrice with PBST and then incubated with HRP-conjugated goat anti-mouse IgG antibody (1:5000; Invitrogen, Waltham, MA, USA) for 1 h at 37 °C. Plates were washed thrice with PBST and developed with 3,3′,5,5′-tetramethylbenzidine (TMB; Sigma-Aldrich, St. Louis, MO, USA) for 15 min at RT. The reaction was stopped by the addition of 2 N H_2_SO_4_ and absorbance was read at 450 nm on a Spark microplate reader (Tecan, Männedorf, Switzerland).

### 2.12. Mouse IFN-γ ELISpot

Splenocytes (5 × 10^5^ cells per well) were incubated in duplicate with NS1.1 or NS1.2 peptide pools (2 µg/mL), or individual NS1-derived peptides (2 µg/mL) in pre-coated 96-well plates (Mouse IFN-γ ELISpot^PLUS^ kit, Mabtech, Nacka Strand, Sweden) for 18 h at 37 °C. Cells incubated with PMA/ionomycin (both Cayman Chemical Company, Ann Arbor, MI, USA) or DMSO were used as positive and negative controls, respectively. Plates were developed according to the manufacturer’s instructions. Developed plates were scanned using an ImmunoSpot S6 Ultimate M2 reader and spots were counted using the ImmunoSpot software version 7.0.9.5 (both Cellular Technology Limited, Shaker Heights, OH, USA). Data are presented as mean SFC per 10^6^ splenocytes after subtraction of the negative control.

### 2.13. Intracellular Cytokine Staining (ICS) and Flow Cytometry

Splenocytes were plated at a density of 2 × 10^6^ cells per well into round-bottom 96-well plates (Sarstedt, Nuembrecht, Germany) and incubated with NS1.1 or NS1.2 peptide pools (2 µg/mL), or individual NS1-derived peptides (2 µg/mL) for a total of 6 h at 37 °C, with brefeldin A (10 µg/mL; Sigma-Aldrich, St. Louis, MO, USA) and monensin (2 µM; BioLegend, San Diego, CA, USA) added for the last 4 h. Cells incubated with PMA/ionomycin or DMSO were used as positive and negative controls, respectively. Cells were harvested, washed with PBS/2% FBS, and then stained with LIVE/DEAD fixable near-IR dead cell stain (Invitrogen, Waltham, MA, USA). Subsequently, cells were incubated with anti-CD16/CD32 (clone 93, Invitrogen, Waltham, MA, USA; 1:500) to block non-specific binding to Fc receptors and surface-stained with anti-CD3e FITC (clone 145-2C11, Invitrogen, Waltham, MA, USA), anti-CD4 PE (clone RM4-5, Invitrogen, Waltham, MA, USA), and anti-CD8a PerCP-Cy5.5 (clone 53-6.7, eBioscience, San Diego, CA, USA). Afterwards, cells were fixed and permeabilised using BD Cytofix/Cytoperm solution (BD Biosciences, San Jose, CA, USA), and then stained with anti-IFN-γ APC (clone XMG1.2, Invitrogen, Waltham, MA, USA). All fluorochrome-labelled antibodies were used at a dilution of 1:200 in BD Brilliant Stain buffer (BD Biosciences, San Jose, CA, USA). Samples were acquired on a BD LSRFortessa X-20 flow cytometer (BD Biosciences, San Jose, CA, USA). Data analysis was performed using FlowJo software version 10.8.2 (BD Biosciences, San Jose, CA, USA). Values obtained under negative control conditions were subtracted from each result.

### 2.14. Statistical Analysis

The non-parametric Mann–Whitney *U* test for unpaired samples was used to compare experimental groups, and *p* values < 0.05 were considered statistically significant. Statistical analyses were carried out using Prism software version 9.5.0 (GraphPad, San Diego, CA, USA).

## 3. Results

### 3.1. Construction and Genetic Characterisation of rMVA-D2-NS1-N207Q

As a proof-of-concept, we used the NS1 coding sequence of the prototype DENV2 strain New Guinea C (NGC) to construct our vaccine candidate. In addition, sequences encoding the second transmembrane domain (TM2) of the E protein (i.e., amino acids 472 to 495), which serves as a natural signal sequence for NS1 [41], were placed upstream of the NS1 coding sequence to ensure proper proteolytic cleavage and targeting of NS1 to the ER lumen. To remove the N-linked glycosylation site at position 207 that was previously shown to be required for NS1-induced endothelial hyperpermeability [28], the corresponding codon was mutated to encode glutamine (i.e., N207Q) (Figure 1a). 

The transgene was subcloned into the vector plasmid pIIIH5redK1L [43], thereby placing it under transcriptional control of the synthetic VACV early/late promoter PmH5 [44], and inserted into deletion site III (del III) of the MVA genome through homologous recombination. The resulting recombinant MVA, rMVA-D2-NS1-N207Q, was repeatedly plaque-purified prior to the preparation of vaccine stocks in primary chicken embryo fibroblasts (CEFs).

PCR analysis of viral genomic DNA confirmed the genetic identity of rMVA-D2-NS1-N207Q (Figure 1b), which was further verified by Sanger sequencing. We also found that rMVA-D2-NS1-N207Q remains genetically stable during serial passage in cell culture (Figure 1c).

### 3.2. Characterisation of Transgene Expression by rMVA-D2-NS1-N207Q

We next sought to characterise rMVA-driven expression of the transgene in cell culture. NS1-N207Q was detected in lysates of rMVA-D2-NS1-N207Q-infected HeLa cells immediately after the viral adsorption step (i.e., 0 hpi), indicating early transcription from the synthetic VACV promoter PmH5, and its levels continued to increase until 24 hpi (Figure 2a), in line with late promoter activity. The protein was not detected in mock-infected cells or cells infected with non-recombinant, wild-type MVA (wtMVA). Furthermore, successful targeting of NS1-N207Q to the ER lumen of A549 cells infected with rMVA-D2-NS1-N207Q was revealed by immunofluorescence co-staining for NS1 and the ER marker calnexin (Figure 2b). Secreted NS1-N207Q appeared in the supernatants of rMVA-D2-NS1-N207Q-infected HeLa cells as early as 2 hpi and further accumulated until 24 hpi (Figure 2c), demonstrating that the protein is rapidly and efficiently secreted.

We then performed additional biochemical characterisation of secreted NS1-N207Q. First, we wanted to assess the conformational state of the protein. We focussed on its dimeric form because secreted hexameric NS1 is highly sensitive to ionic detergents [45,46] and thus not detectable under the conditions of SDS-PAGE. NS1 dimers are known to dissociate into monomers upon heat denaturation [15]. Therefore, supernatant samples of rMVA-D2-NS1-N207Q-infected HeLa cells were either boiled or not boiled prior to SDS-PAGE and Western blotting. In heat-denatured samples, NS1-N207Q was present as a monomer, whereas it was found as a dimer in samples that were not exposed to heat (Figure 2d). This shows that NS1-N207Q is secreted at least as dimers into the extracellular milieu. Next, we sought to prove the lack of N-linked glycosylation at position 207 of secreted NS1-N207Q. To this end, supernatant samples of HeLa cells infected with rMVA-D2-NS1-N207Q were digested with endoglycosidase H (Endo H; removes high-mannose glycans [47]) or peptide:N-glycosidase F (PNGase F; removes high-mannose, complex and hybrid glycans [48]) and subsequently analysed by Western blotting. NS1-N207Q displayed increased electrophoretic mobility following PNGase F digestion, indicating removal of the complex glycan at residue 130, whereas no mobility shift was observed upon treatment with Endo H (Figure 2e), thus confirming the absence of the high-mannose glycan at residue 207.

### 3.3. Immunogenicity of rMVA-D2-NS1-N207Q in Mice

To assess the immunogenicity of rMVA-D2-NS1-N207Q, we examined NS1-specific antibody and T-cell responses in C57BL/6J mice. Groups of mice were immunised twice intramuscularly on days 0 and 28 with 10^7^ plaque forming units (PFU) of rMVA-D2-NS1-N207Q or wtMVA (empty vector control) (Figure 3a). Sera were collected on days 0 (pre-immune), 28 (pre-boost), and 56 (post-boost), and tested for NS1-specific antibodies by enzyme-linked immunosorbent assay (ELISA). We systemically screened the sera for antibodies binding to different conformations of NS1 by coating plates with (i) native recombinant NS1 (i.e., hexamer), (ii) detergent-treated NS1 (i.e., dimer) [45,46], or (iii) detergent-treated, heat-denatured NS1 (i.e., monomer) [15]. High levels of NS1 hexamer–binding antibodies were already detected in the sera of rMVA-D2-NS1-N207Q-immunised mice collected after a single immunisation, but did not further increase after the second immunisation (Figure 3b). The same pattern was observed for antibodies binding to NS1 dimers (Figure 3c). Interestingly, much lower reactivity with heat-denatured NS1 was observed for sera collected on day 28 (Figure 3d), suggesting that a large proportion of NS1-specific antibodies recognised conformational epitopes. However, boosting strongly increased the levels of NS1 monomer–binding antibodies. These data indicate that a robust NS1-specific antibody response was induced by prime–boost vaccination with rMVA-D2-NS1-N207Q.

Spleens were harvested on day 56 and splenocytes were restimulated with NS1-derived peptides, after which antigen-specific T-cell responses were measured by IFN-γ ELISpot and flow cytometry. IFN-γ–producing T cells were more abundant in the spleens of rMVA-D2-NS1-N207Q-immunised mice compared with wtMVA-immunised mice and responded to both NS1.1 and NS1.2 peptide pools, which span the N-terminal half and C-terminal half of NS1, respectively, though the response to the former was more pronounced (Figure 4a). We found that the strong response to the NS1.1 pool was largely accounted for by peptides p9 and p19/p20, which cover immunodominant NS1 epitopes previously identified in both C57BL/6 and BALB/c mice [49] (Figure 4b). Further characterisation of the T cells induced after rMVA-D2-NS1-N207Q immunisation by flow cytometry showed that IFN-γ^+^ cells responding to the NS1.1 pool were of the CD4^+^ T-cell subset (Figure 4c,d). The subset of T cells responding to NS1.2 could not be identified because their frequency was too low (Figure 4c,d). In summary, we conclude that rMVA-D2-NS1-N207Q is highly immunogenic in mice, inducing antibodies that bind all major NS1 conformations as well as NS1-specific IFN-γ–producing CD4^+^ T cells.

## 4. Discussion

NS1 is highly conserved among DENV1–4, and cross-reactive NS1-specific antibodies have been shown to protect against lethal vascular leakage caused by heterologous serotypes [24,29]. It has thus been proposed that vaccines delivering NS1 of a single serotype might be sufficient to confer significant protection against multiple serotypes. However, a potential drawback of using NS1 as a vaccine antigen is its intrinsic ability to trigger vascular leakage in vivo. Removal of the N-linked glycosylation site at position 207 was previously shown to abrogate NS1-induced endothelial hyperpermeability [28]. Here, we used this information to construct a DENV2 NS1–based vaccine candidate (rMVA-D2-NS1-N207Q) that would be safe to administer.

In an earlier study, the NS1-N207Q mutant was less efficiently secreted and exhibited lower stability in the supernatant than its wild-type counterpart [50], whereas more recent work observed no effects of the N207Q substitution on NS1 secretion and stability [28]. Both reports also showed that NS1-N207Q was secreted as a hexamer. We found no evidence of increased intracellular accumulation of NS1-N207Q in rMVA-D2-NS1-N207Q-infected cells, which would have been indicative of defective secretion, but observed high levels of secreted protein in the supernatant. Though we only demonstrated the presence of dimeric subunits of NS1-N207Q in the supernatant of rMVA-D2-NS1-N207Q-infected cells, it is conceivable that the secreted form is in fact a hexamer, based on the findings of the above studies.

We showed that rMVA-D2-NS1-N207Q induced robust NS1-specific antibody responses in mice and characterised these responses further by measuring antibody reactivity to different conformations of NS1, though we cannot confirm that these oligomeric states of NS1 were stably maintained after coating onto ELISA plates. Most NS1 B-cell epitopes described to date are linear [51,52], but several studies have observed significant reductions in antibody binding upon heat denaturation of NS1, suggesting the presence of conformational epitopes [53,54,55,56,57]. In agreement with these studies, we found that pre-boost sera of rMVA-D2-NS1-N207Q-immunised mice exhibited reduced reactivity with monomeric heat-denatured NS1 compared with dimeric or hexameric NS1. However, this difference was no longer seen for post-boost sera, indicating that the antibody response to linear epitopes on NS1 strongly increased with the booster immunisation. We do not anticipate the N207Q substitution to have an impact on the recognition of NS1 by B cells, as the majority of B-cell epitopes identified in humans and animal models is located within different regions of NS1 [51,52].

Largely unexplored has been the presence of NS1 B-cell epitopes that span the two monomers within a dimer or neighbouring dimers within a hexamer. Such quaternary epitopes have previously been found within and across E protein dimers present on intact dengue virions [58,59,60], but are yet to be reported for NS1 oligomers. Using sera of rMVA-D2-NS1-N207Q-immunised mice, we did not observe any difference between the levels of antibodies binding dimeric or hexameric NS1, indicating the absence of quaternary epitopes spanning adjacent NS1 dimers. Our data suggest that most antibody binding sites are present on the dimeric form of NS1, with residues from one or both protomer(s) likely participating in the contacts.

An increasing number of studies supports a role for T cells in protecting from severe dengue disease [61,62,63,64,65,66]. In humans, NS1 is a target of both CD4^+^ and CD8^+^ T cells, though to a lesser extent than other non-structural proteins such as NS3 and NS5 [67,68,69,70], and some NS1 T-cell epitopes identified in mice overlap with those found in human subjects [49,71]. Earlier work demonstrated that NS1-specific CD4^+^ T cells induced by vaccination were critical for protection against lethal DENV2 infection in BALB/c mice, but that this was independent of their helper functions [72]. Interestingly, NS1-specific CD8^+^ T cells were only partially protective in that model [72]. Another study showed that T cells induced by ZIKV NS1 vaccination were essential to limiting ZIKV viraemia in BALB/c mice [73]. We observed that rMVA-D2-NS1-N207Q elicited NS1-specific CD4^+^, but not CD8^+^, T-cell responses in C57BL/6J mice (Figure 4). The majority of this response was directed to two immunodominant epitopes located in the N-terminal half of NS1 that were previously identified in NS1-vaccinated BALB/c and C57BL/6 mice [49], thus confirming these findings. 

## 5. Conclusions

Collectively, the data presented here support rMVA-D2-NS1-N207Q as a promising and potentially safer alternative to existing NS1-based vaccine candidates, capable of inducing high levels of NS1-specific antibodies as well as NS1-specific T cells. Further assessment of the protective efficacy of rMVA-D2-NS1-N207Q in a relevant mouse model of DENV infection seems warranted.

## Figures and Tables

**Figure 1 vaccines-11-00714-f001:**
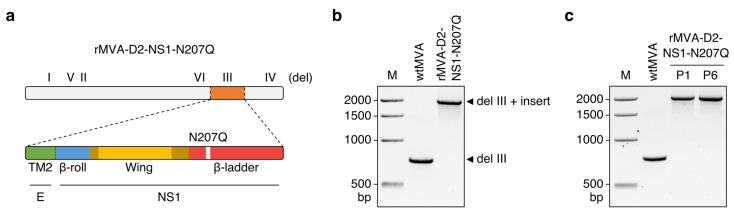
Construction and genetic characterisation of rMVA-D2-NS1-N207Q. (**a**) Schematic representation of rMVA-D2-NS1-N207Q. Top shows the approximate location of the transgene (orange) and the major deletion sites (del) I–VI in the MVA genome. Bottom shows the properties of the transgene product. The N-terminal β-roll is coloured blue (amino acids 1 to 29), the α/β subdomain of the wing domain is depicted in yellow (aa 38–151), connector subdomains within the wing domain are shown in gold (aa 30–37 and 152–180), and the C-terminal β-ladder is coloured red (aa 181–352). The N207Q substitution is highlighted in white. The second transmembrane domain (TM2) of the envelope (E) protein is shown in green. The dotted line between TM2 and β-roll indicates post-translational cleavage. (**b**) Viral genomic DNA was extracted from stocks of non-recombinant, wild-type MVA (wtMVA), and rMVA-D2-NS1-N207Q, and analysed by del III–specific PCR. M, 1 kb DNA ladder. (**c**) rMVA-D2-NS1-N207Q stock (passage 1 [P1]) was passaged five times in CEFs. Viral genomic DNA was extracted from P1 and P6 viruses, and analysed by del III–specific PCR. M, 1 kb DNA ladder.

**Figure 2 vaccines-11-00714-f002:**
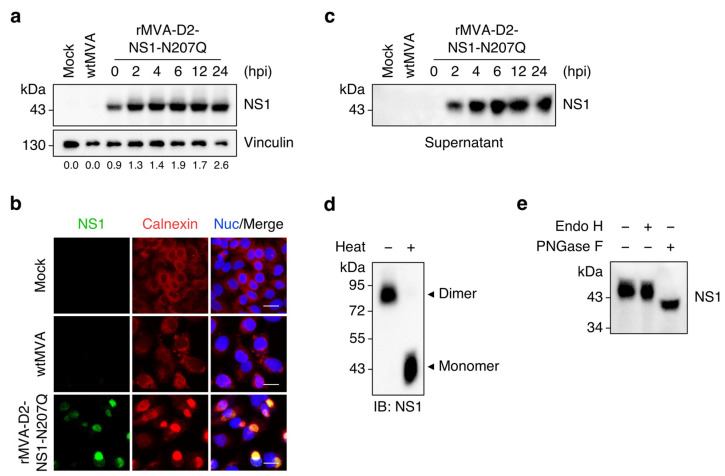
Characterisation of transgene expression by rMVA-D2-NS1-N207Q. (**a**) HeLa cells were mock-infected or infected with wtMVA or rMVA-D2-NS1-N207Q (MOI = 3) and lysed at 24 hpi (mock and wtMVA) or at the indicated time points (rMVA-D2-NS1-N207Q). Whole-cell lysates were analysed by Western blotting with antibodies against DENV NS1 and vinculin (loading control). Relative NS1 levels were determined by densitometric analysis. Note that the 0 hpi time point was collected immediately after 1 h of virus adsorption. (**b**) A549 cells were mock-infected or infected with wtMVA or rMVA-D2-NS1-N207Q (MOI = 3). At 24 hpi, cells were analysed by immunofluorescence staining for DENV NS1 (green) and calnexin (red). Nuclei were stained with Hoechst 33342 (blue). Scale bars, 20 µm. (**c**–**e**) HeLa cells were mock-infected or infected with wtMVA or rMVA-D2-NS1-N207Q (MOI = 3). (**c**) Supernatant samples were collected at 24 hpi (mock and wtMVA) or at the indicated time points (rMVA-D2-NS1-N207Q) and analysed by Western blotting with an antibody against DENV NS1. Note that the 0 hpi time point was collected immediately after 1 h of virus adsorption. (**d**) Supernatant samples from cells infected with rMVA-D2-NS1-N207Q were collected at 24 hpi, boiled (or not) for 5 min at 95 °C prior to non-reducing SDS-PAGE, and analysed by Western blotting with an antibody against DENV NS1. (**e**) Supernatant samples from cells infected with rMVA-D2-NS1-N207Q were collected at 24 hpi, left untreated or digested with Endo H or PNGase F, and analysed by Western blotting with an antibody against DENV NS1.

**Figure 3 vaccines-11-00714-f003:**
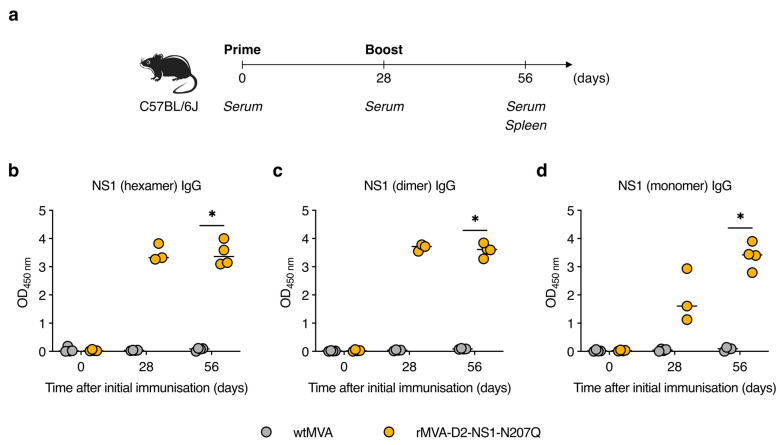
rMVA-D2-NS1-N207Q induces robust NS1-specific antibody responses in mice. (**a**) Experimental protocol. C57BL/6J mice were immunised intramuscularly with 10^7^ PFU of wtMVA (*n* = 4) or rMVA-D2-NS1-N207Q (*n* = 4) on days 0 and 28. (**b**–**d**) Sera collected on days 0, 28, and 56 were diluted 100-fold and tested for NS1-specific IgG antibodies by ELISA using plates coated with NS1 hexamers (**b**), NS1 dimers (**c**), or NS1 monomers (**d**). Note that sera of only three mice from the rMVA-D2-NS1-N207Q group were obtained on day 28. Each data point represents one mouse and short horizontal lines represent medians. *, *p* < 0.05 (Mann–Whitney *U* test).

**Figure 4 vaccines-11-00714-f004:**
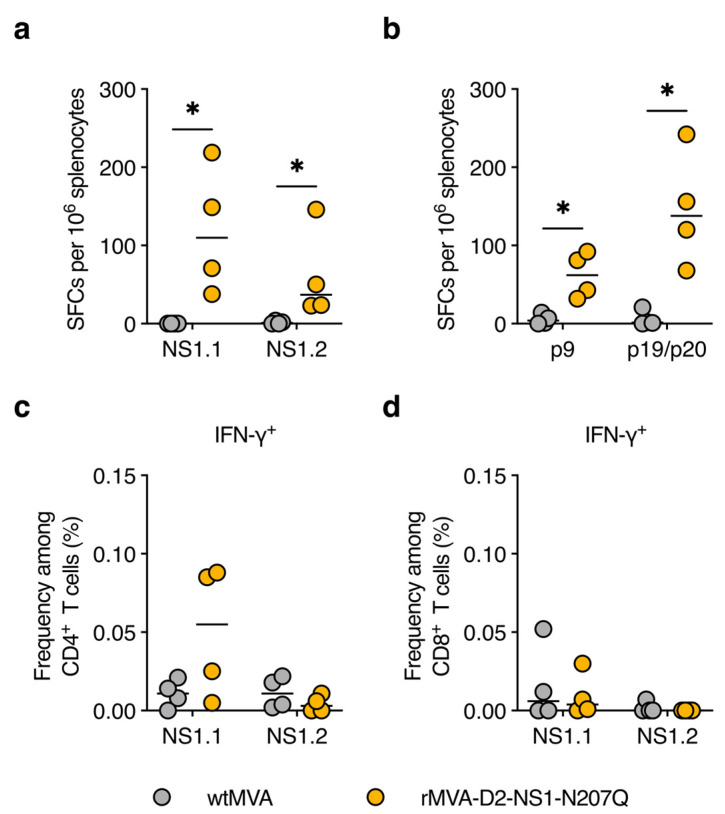
rMVA-D2-NS1-N207Q induces NS1-specific CD4^+^ T-cell responses in mice. Same mouse experiment as in Figure 3. Spleens were harvested on day 56 and splenocytes were restimulated with overlapping peptide pools NS1.1 or NS1.2 (**a**,**c**,**d**), or individual NS1-derived peptides covering known immunodominant epitopes (**b**). (**a**,**b**) Numbers of IFN-γ–producing cells were quantified by ELISpot. Data were DMSO background–subtracted and are expressed as spot-forming cells (SFCs) per 10^6^ splenocytes. (**c**,**d**) Frequencies of IFN-γ–producing CD4^+^ T cells (**c**) and CD8^+^ T cells (**d**) were measured by flow cytometry. Each data point represents one mouse and short horizontal lines represent medians. *, *p* < 0.05 (Mann–Whitney *U* test).

## Data Availability

Not applicable.

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
