# Peer review of "Recombinant Modified Vaccinia Virus Ankara Expressing a Glycosylation Mutant of Dengue Virus NS1 Induces Specific Antibody and T-Cell Responses in Mice"

_vaccines, 2023, doi:10.3390/vaccines11040714_

Round 1
Reviewer 1 Report
The authors report a MVA construct that expresses a partially de-glycosylated NS1 protein at N209 site, with the rationale that WT NS1 may not be safe since it has been shown to interact with endothelial cells and this process involves the glycans at N209.
The MVA-NS1N207Q vaccine candidate was found to be immunogenic in immune competent C57bl/6 mice in a homologous prime-boost regimen. NS1-specific antibodies were detected in the immune sera and splenocytes responded to re-stimulation with NS1 peptides.
The NS1-specific antibody response could have been further characterized against NS1 proteins from other DENV serotypes. In addition, the obvious missing part is the challenge experiment to evaluate the protective efficacy of this vaccine candidate. Finally, it would have been interesting to compare MVA constructs expressing either WT or deglycosylated NS1, in terms of immunogenicity and safety (any signs of endothelial leakage with the WT construct?).
Please see my specific comments below.
Specific comments:
-Abstract and Introduction: Please update by including the latest live attenuated tetravalent vaccine developed by Takeda that allows immunization of dengue naïve individuals. This vaccine has obtained approval in Indonesia and Europe.
-The role of NS1 in dengue pathogenesis in mouse models, and hence its value as vaccine candidate remains controversial and may depend on the DENV strain that is employed to conduct the studies. Please refer to this study https://doi.org/10.1084/jem.20191548
-Figure 2b: it is not clear whether confocal microscopy was performed here. Typically, confocal microscopy would be required in order to confirm co-localization of NS1 and calnexin signals. Alternatively, proximity ligation assay (PLA) could be performed as it does not require a confocal microscope.
-The authors may want to include an ethics statement regarding the cryopreserved human PBMCs used in this study.
-Figure 2d: The production of hexameric NS1 in the culture supernatant could have been assessed in a non-denaturing non-reducing electrophoresis approach. This is particularly relevant since protective immunity is expected to target hexameric NS1.
-Figure 2f: the positive and negative controls PMA /Ionomycin and DMSO respectively, should be included. Also, could part of the T cell response observed be driven by E protein TM2 that was inserted in the construct?
-Figure 3: It would be good to show that the various NS1 suspensions used to coat the ELISA plates display the expected conformation (monomeric, dimeric and hexameric). This aspect is particularly important because the authors discuss quite a bit the implication of inducing antibodies that recognize linear or conformational B cell epitopes in NS1 protein.
-Figure 4: A number of controls are missing: a naïve non-immunized group (baseline for non-specific reactivity); PMA/Ionomycin and DMSO control for panels a&b;
Author Response
Reviewer 1
The authors report a MVA construct that expresses a partially de-glycosylated NS1 protein at N209 site, with the rationale that WT NS1 may not be safe since it has been shown to interact with endothelial cells and this process involves the glycans at N209.
The MVA-NS1N207Q vaccine candidate was found to be immunogenic in immune competent C57bl/6 mice in a homologous prime-boost regimen. NS1-specific antibodies were detected in the immune sera and splenocytes responded to re-stimulation with NS1 peptides.
The NS1-specific antibody response could have been further characterized against NS1 proteins from other DENV serotypes. In addition, the obvious missing part is the challenge experiment to evaluate the protective efficacy of this vaccine candidate. Finally, it would have been interesting to compare MVA constructs expressing either WT or deglycosylated NS1, in terms of immunogenicity and safety (any signs of endothelial leakage with the WT construct?).
Please see my specific comments below.
Specific comments:
-Abstract and Introduction: Please update by including the latest live attenuated tetravalent vaccine developed by Takeda that allows immunization of dengue naïve individuals. This vaccine has obtained approval in Indonesia and Europe.
We thank the reviewer for making us aware of this and have modified the text accordingly
-The role of NS1 in dengue pathogenesis in mouse models, and hence its value as vaccine candidate remains controversial and may depend on the DENV strain that is employed to conduct the studies. Please refer to this study https://doi.org/10.1084/jem.20191548
We agree and have cited this study in the revised version of the manuscript
-Figure 2b: it is not clear whether confocal microscopy was performed here. Typically, confocal microscopy would be required in order to confirm co-localization of NS1 and calnexin signals. Alternatively, proximity ligation assay (PLA) could be performed as it does not require a confocal microscope.
We have used widefield fluorescence microscopy and agree that this does not per se confirm co-localization; this term was therefore removed from the text. Calnexin was simply used as an ER marker in order to prove localization of NS1 to the ER, not to specifically prove co-localization of NS1 and calnexin. We have rewritten this section accordingly (lines 524–527).
-The authors may want to include an ethics statement regarding the cryopreserved human PBMCs used in this study.
Based on the comments of one of the other reviewers, we have decided to leave out Figure 2f because the results were considered preliminary.
-Figure 2d: The production of hexameric NS1 in the culture supernatant could have been assessed in a non-denaturing non-reducing electrophoresis approach. This is particularly relevant since protective immunity is expected to target hexameric NS1.
We agree with the reviewer that this would have been the optimal approach. Yet, the detection of dimeric NS1 subunits in the culture supernatant strongly suggests that they are part of secreted hexameric NS1, as the dimeric form itself is not secreted.
-Figure 2f: the positive and negative controls PMA /Ionomycin and DMSO respectively, should be included. Also, could part of the T cell response observed be driven by E protein TM2 that was inserted in the construct?
Based on the comments of one of the other reviewers, we have decided to leave out Figure 2f because the results were considered preliminary.
-Figure 3: It would be good to show that the various NS1 suspensions used to coat the ELISA plates display the expected conformation (monomeric, dimeric and hexameric). This aspect is particularly important because the authors discuss quite a bit the implication of inducing antibodies that recognize linear or conformational B cell epitopes in NS1 protein.
It has already been shown by others that hexameric NS1 can be separated into dimers and monomers through treatment with detergents and exposure to heat, respectively. We therefore believe that further confirmation is not necessary. We have referred to the respective studies in lines 600–601.
-Figure 4: A number of controls are missing: a naïve non-immunized group (baseline for non-specific reactivity); PMA/Ionomycin and DMSO control for panels a&b
We did not include a naïve non-immunized group as we have never seen responses to virus-derived peptide pools in historic control groups (e.g., mice injected with PBS or TBS) and because we had to comply with the 3R principle, which demands that the number of experimental animals should be reduced to the necessary minimum.
DMSO and PMA/ ionomycin controls were included in the experiments. DMSO gave rise to very low numbers of spot forming cells, which were subtracted from the numbers measured with peptides (this was already indicated in the materials and methods and has now also been added to the caption of Figure 4). PMA/ionomycin was included as positive control for the functional integrity of the splenocytes and gave rise to many SFCs, in most cases too many to count.
Reviewer 2 Report
Thr article title "Recombinant Modified Vaccinia Virus Ankara Expressing a Glycosylation Mutant of Dengue Virus NS1 Induces Specific Antibody and T-Cell Responses in Mice" by Lucas Wilken et al. is interesting for novel exploration vaccine candidate for dengue virus.
There are some minor comments:
1. I wonder whether only NS1 from dengue serotype is enough for overall protection of dengue virus infection if it is applied in humans.
2. Is it important to choose the representative viral sequence from dengue serotypes for candidate vaccine? Do we need the sequence of severe dengue virus as a sequence model?
3. Do you think other glycosylation sie of NS1 not play role in pathogenesis of dengue infection?
Author Response
Reviewer 2
Thr article title "Recombinant Modified Vaccinia Virus Ankara Expressing a Glycosylation Mutant of Dengue Virus NS1 Induces Specific Antibody and T-Cell Responses in Mice" by Lucas Wilken et al. is interesting for novel exploration vaccine candidate for dengue virus.
There are some minor comments:
- I wonder whether only NS1 from dengue serotype is enough for overall protection of dengue virus infection if it is applied in humans.
Given the high sequence similarity (65–75%) between the NS1s of the four DENV serotypes, significant antibody and T-cell cross-reactivity is expected. Others have already shown that vaccination with NS1 of DENV1,3, or 4 can induce partial protection against DENV2 NS1-induced endothelial hyperpermeability and DENV2 infection in mice (see https://www.science.org/doi/10.1126/scitranslmed.aaa3787). A vaccine candidate based on the NS1 of a single DENV serotype might therefore be sufficient; however, this needs to be further investigated.
- Is it important to choose the representative viral sequence from dengue serotypes for candidate vaccine? Do we need the sequence of severe dengue virus as a sequence model?
We kindly ask the reviewer to refer to our answer to their first comment.
- Do you think other glycosylation sie of NS1 not play role in pathogenesis of dengue infection?
The other glycosylation site (N130) is apparently not involved in NS1-induced endothelial hyperpermeability (see https://dx.plos.org/10.1371/journal.ppat.1007938).
Reviewer 3 Report
In this manuscript entitled “Recombinant Modified Vaccinia Virus Ankara Expressing a Glycosylation Mutant of Dengue Virus NS1 Induces Specific Antibody and T-Cell Responses in Mice” by Wilken and colleagues, the authors used a modified DENV2 NS1 (N207Q) inserted into a modified vaccinia virus Ankara (MVA) used as a viral vector (rMVA-D2-NS1-N207Q) to immunize mice. The authors showed that rMVA-D2-NS1-N207Q is capable of inducing antibody responses and activating NS1-specific T cells in mice. These findings support the development of potentially safer vaccines alternative to the existing NS1-based vaccine candidates. However, there are several issues that need to be addressed in order to consider its publication as following depicted:
Major(s)
1. The main topic of the title of this manuscript is the use of the modified vaccinia Ankara virus to express the NS1 protein of DENV, however, no mention of the modified vaccinia Ankara virus as a experimental viral vector is included in the introduction. I think the authors should mention it why this is an important new vector for non-poxvirus diseases.
2. For plasmid construction and vector insertion of the DENV NS1 nucleotide sequence, it seems that the authors only used the DENV2 stain New Guinea C sequence for NS1 to construct their vaccine candidate. Given the importance that any new DENV vaccine candidate must generate a highly cross-reactive and protective immune response against all four serotypes, the authors must justify why using the sequence of only one DENV serotype and not all of them or a consensus sequence?
3. Experiments to examine the ability of this vaccine candidate to activate human NS1-specific T cell in vitro are not convincing as results from only one donor were included. Besides the results of this single experiment seem inconclusive. The authors must increase the numbers of donors to reach a robust conclusion about whether the NS1-N207Q was properly processed or presented by APCs leading to the activation of NS1-specific T cells.
4. Additionally, please clarify why a wtMVA should induce a T cell response even higher than the two NS1 peptide pools in a DENV-pre-immune donor?
5. Also, was any normalization performed in this experiment? For instance, number of infected cells, amount of NS1-input (e.g., hexamer), peptides? Were any experimental duplicates included? Please clarify this.
6. In the in vivo mouse experiments when evaluating the immunogenicity of MVA-D2-NS1-N207Q, did authors measured the amount of secreted NS1 after infecting the mice?
7. In the same line, there was not indication of NS1-induced adverse effects detected in the immunized mice?
8. In Figure 3, and this is very important to be considered; even though the NS1 hexamer or any its oligomeric states should break into less complex conformational states such as dimer or monomers after treatment with detergents or boiling - as the authors mentioned - once these NS1 proteins are used to coat the surface of ELISA plates, it is very difficult to maintain them as single dimers or monomers. Therefore, the authors must be careful in concluding about whether the antibody responses against NS1 were primarily NS1-conformational dependent in immunized mice or not. Point taken; immunization did elicit an NS1-antibody responses in mice. Please revise these results and their conclusions.
9. Additionally, was any molar ratio of the three “NS1 conformational states” hexamers, dimers and monomers adjusted while coating the ELISA plates? Please clarify this point.
10. In Figure 3, what is represented in the graphs are overall absorbance values, end-point dilution titers? In the methods section, the authors said, line 236 “Antibody endpoint titers were determined…”. Please clarify this point as it is critical for final conclusions.
11. In the same line, was any comparison done between mice immunization with a rMVA-wtNS1 vs rMVA-D2-NS1-N207Q mutant?
12.
Minor(s)
1. Line 63 of the Introduction section. Please adjust reference according to the Journal. “Pryor and Wright, 1994”.
2. Please indicate in the main text which cell line was used to characterize the transgene expression of the rMVA-D2-NS1-N207Q and the production of both wildtype and N207Q NS1 mutant?
Author Response
Reviewer 3
In this manuscript entitled “Recombinant Modified Vaccinia Virus Ankara Expressing a Glycosylation Mutant of Dengue Virus NS1 Induces Specific Antibody and T-Cell Responses in Mice” by Wilken and colleagues, the authors used a modified DENV2 NS1 (N207Q) inserted into a modified vaccinia virus Ankara (MVA) used as a viral vector (rMVA-D2-NS1-N207Q) to immunize mice. The authors showed that rMVA-D2-NS1-N207Q is capable of inducing antibody responses and activating NS1-specific T cells in mice. These findings support the development of potentially safer vaccines alternative to the existing NS1-based vaccine candidates. However, there are several issues that need to be addressed in order to consider its publication as following depicted:
Major(s)
- The main topic of the title of this manuscript is the use of the modified vaccinia Ankara virus to express the NS1 protein of DENV, however, no mention of the modified vaccinia Ankara virus as a experimental viral vector is included in the introduction. I think the authors should mention it why this is an important new vector for non-poxvirus diseases.
As the reviewer suggested we have included an additional paragraph to introduce MVA as a promising viral vector for the delivery of non-poxvirus viral antigens (lines 89–94)
- For plasmid construction and vector insertion of the DENV NS1 nucleotide sequence, it seems that the authors only used the DENV2 stain New Guinea C sequence for NS1 to construct their vaccine candidate. Given the importance that any new DENV vaccine candidate must generate a highly cross-reactive and protective immune response against all four serotypes, the authors must justify why using the sequence of only one DENV serotype and not all of them or a consensus sequence?
We agree that novel vaccines must induce cross protective immunity to all four serotypes if DENV. The experiments presented in the present manuscript should be considered “proof of principle” to show that is possible to induce NS1-specific antibody and T-cell responses with a recombinant MVA. Considering the high sequence identity across NS1 of the respective serotypes (65–75%) we expect sufficient cross-reactivity with NS1 proteins of heterologous serotypes. Others have already shown that vaccination with NS1 of DENV1,3, or 4 can induce partial protection against DENV2 NS1-induced endothelial hyperpermeability and DENV2 infection in mice (see https://www.science.org/doi/10.1126/scitranslmed.aaa3787).
- Experiments to examine the ability of this vaccine candidate to activate human NS1-specific T cell in vitro are not convincing as results from only one donor were included. Besides the results of this single experiment seem inconclusive. The authors must increase the numbers of donors to reach a robust conclusion about whether the NS1-N207Q was properly processed or presented by APCs leading to the activation of NS1-specific T cells.
We agree with the reviewer that the data obtained with PBMC of DENV exposed human subjects is rather preliminary and we therefore have decided to take it out. The data obtained after immunization of mice are more compelling and show that the construct is immunogenic and able to induce NS1 specific T cell responses. Apparently upon vaccination, NS1 protein is expressed, processed and presented leading to the induction of specific T cells in vivo.
- Additionally, please clarify why a wtMVA should induce a T cell response even higher than the two NS1 peptide pools in a DENV-pre-immune donor?
As indicated above, we have taken out the preliminary in vitro T cell data
- Also, was any normalization performed in this experiment? For instance, number of infected cells, amount of NS1-input (e.g., hexamer), peptides? Were any experimental duplicates included? Please clarify this.
As indicated above, we have taken out the preliminary in vitro T cell data
- In the in vivo mouse experiments when evaluating the immunogenicity of MVA-D2-NS1-N207Q, did authors measured the amount of secreted NS1 after infecting the mice?
No, this was not done. We focused on the induction of NS1-specific antibody and T-cell responses.
- In the same line, there was no indication of NS1-induced adverse effects detected in the immunized mice?
No, vaccination with MVA-D2-NS1-N207Q was well tolerated and the vaccinated mice did not show any clinical signs.
- In Figure 3, and this is very important to be considered; even though the NS1 hexamer or any its oligomeric states should break into less complex conformational states such as dimer or monomers after treatment with detergents or boiling - as the authors mentioned - once these NS1 proteins are used to coat the surface of ELISA plates, it is very difficult to maintain them as single dimers or monomers. Therefore, the authors must be careful in concluding about whether the antibody responses against NS1 were primarily NS1-conformational dependent in immunized mice or not. Point taken; immunization did elicit an NS1-antibody responses in mice. Please revise these results and their conclusions.
We thank the reviewer for raising this point. We have addressed this limitation in the Discussion section (lines 1154–1155).
- Additionally, was any molar ratio of the three “NS1 conformational states” hexamers, dimers and monomers adjusted while coating the ELISA plates? Please clarify this point.
No, the molar ratios were not adjusted. All wells were coated with 1 µg/mL of recombinant protein irrespective of the conformational state.
- In Figure 3, what is represented in the graphs are overall absorbance values, end-point dilution titers? In the methods section, the authors said, line 236 “Antibody endpoint titers were determined…”. Please clarify this point as it is critical for final conclusions.
Figures 3b–d showed OD450nm absorbance values for 100-fold diluted sera collected at 0, 28 and 56 dpi; Figure 3e showed endpoint titers for sera collected on day 56, thereby providing a measure for the quantity of NS1-specific antibodies induced after two immunizations. We have removed Figure 3e to prevent any confusion and because we feel that the data shown in Figures 3b–d already provide sufficient information on NS1-specific antibody responses.
- In the same line, was any comparison done between mice immunization with a rMVA-wtNS1 vs rMVA-D2-NS1-N207Q mutant?
No, an rMVA expressing wild-type NS1 was not constructed and therefore no such comparison was performed.
Minor(s)
- Line 63 of the Introduction section. Please adjust reference according to the Journal. “Pryor and Wright, 1994”.
This has been corrected
- Please indicate in the main text which cell line was used to characterize the transgene expression of the rMVA-D2-NS1-N207Q and the production of both wildtype and N207Q NS1 mutant?
Characterization of transgene expression was performed in HeLa and A549 cells. The latter were used specifically for immunofluorescence staining, as their morphology was less affected by MVA infection than that of HeLa cells. Virus stocks were prepared in chicken embryo fibroblasts (CEFs). This information has now been provided in the main text.
Round 2
Reviewer 1 Report
I maintain that it is essential to include a control group of naive mice for Figure 4. The spirit of the 3R rule is certainly not meant to omit important controls.
Confirming the conformation of hexameric, dimeric and monomeric forms of NS1 is also rather critical to validate the interpretation of the ELISA findings. The fact that others have shown it in other studies is not sufficient, unless these proteins were actually made by the same labs and/or unless these NS1 proteins were purchased from a commercial source that provides information on their products.
Author Response
I maintain that it is essential to include a control group of naive mice for Figure 4. The spirit of the 3R rule is certainly not meant to omit important controls.
We agree with the reviewer in case the study would have included vaccination challenge experiments in which mice are infected with DENV after vaccination. Then, we agree, it would be crucial to include PBS control mice to exclude non-specific immune responses that could interfere with the replication of the challenge virus. However, in our study we assessed the immunogenicity of the MVA-NS1 construct only. We could exclude that the vector backbone induced DENV-specific T cell responses. Inclusion of an extra control group would not have changed that conclusion.
Confirming the conformation of hexameric, dimeric and monomeric forms of NS1 is also rather critical to validate the interpretation of the ELISA findings. The fact that others have shown it in other studies is not sufficient, unless these proteins were actually made by the same labs and/or unless these NS1 proteins were purchased from a commercial source that provides information on their products
We agree with the reviewer that it is important to know what the antibodies raised after immunization with rMVA-D2-NS1-N207Q recognize. We attempted to address this by using NS1 protein that was denatured in different ways. Using these antigens for coating of the ELISA plates we observed a difference in antibody reactivity between native recombinant NS1 or detergent-treated NS1 on the one hand and detergent-treated, heat-denatured NS1 on the other. Our interpretation was that the antibody response to the latter, which most likely represents NS1 in its monomeric form (as was shown by others), is weaker than against NS1 in a higher organization, despite not having shown the oligomeric state of the protein. Although this could be considered a limitation of the study, the conclusion of these experiments that a proportion of rMVA-D2-NS1-N207Q induced antibodies is directed to conformational epitopes and this could be observed after a single immunization only, seems justified.
Reviewer 3 Report
The authors have thorgoughly addressed all concerns reaised by this reviewer.
Author Response
We are pleased that the reviewer agreed with our response to his comments